# Enhanced Online Segmentation and Performance Evaluation Method for Real-Time Activity Recognition in Smart Homes

Abbas Ramadan
*Lab-STICC, UMR CNRS 6285*
*University of South Brittany*
Lorient, France
abbas.ramadan@univ-ubs.fr

Farida Saïd
*LMBA, UMR CNRS 6205*
*University of South Brittany*
Lorient, France
farida.said@univ-ubs.fr

Florent Frizon De Lamotte
*Lab-STICC, UMR CNRS 6285*
*University of South Brittany*
Lorient, France
florent.lamotte@univ-ubs.fr

*Abstract*—This paper presents a robust method for real-time recognition of Activities of Daily Living (ADLs) in smart home environments using IoT data. Our approach improves the segmentation of sensor data streams into distinct activities by leveraging IoT sensor spatiotemporal features and applies the Needleman-Wunsch method to align predicted and actual activities. Testing on the Aruba dataset achieved 83.2% accuracy, demonstrating superior performance in segmentation and activity recognition compared to existing dynamic methods. Future work will focus on developing a sensor installation simulator to enhance accuracy and reliability.

*Index Terms*—Activity of Daily Living, ADLs, IoT Data, Human Activity Recognition, Smart Home, Real-time Activity Recognition, Spatiotemporal Segmentation, Dynamic Segmentation Algorithms.

## I. Introduction

The rapid advancement of ambient computing has led to the development of various sensors and devices within assistive environments, particularly for elderly or disabled individuals. These Ambient Assisted Living (AAL) systems integrate wireless networks, sensors, and software applications to provide health monitoring, personal assistance, and telemedicine services [1], enabling individuals to maintain their independence at home.

A key component of these systems is the ability to monitor and analyze daily activities using data from connected devices. This data provides valuable insights into individuals' behavior, habits, and routines, helping to assess their health, detect early signs of vulnerability, and prevent emergencies. However, variability in how individuals perform daily activities can lead to inconsistencies in the collected data, making it challenging to segment sensor data into distinct activities and assign accurate labels.

To address these challenges, we propose leveraging spatiotemporal correlations between sensors to enhance segmentation accuracy, particularly in real-time recognition of Activities of Daily Living (ADLs) [2], [3]. In this context, technologies such as artificial intelligence (AI), the Internet of Things (IoT), and *digital twins* play a crucial role. These technologies have significantly improved activity recognition in smart homes by providing more reliable data processing and simulation capabilities [4]. *Digital twins*, in particular, offer virtual representations of smart home environments, simulating system functionality and aiding in monitoring and predictive analysis [5].

This study focuses on improving spatiotemporal correlation parameters for more accurate real-time dynamic segmentation of sensor data. Additionally, we propose methods to evaluate recognition performance when predicted and actual activity sequences do not perfectly align.

This paper is structured as follows: Section II presents the background and challenges related to ADL recognition. Section III details our methodology. The results of our experiments are discussed in Section V, followed by concluding remarks and future research directions in Section VI.

## II. Background

Activities of Daily Living (ADLs) encompass essential tasks required for maintaining personal well-being, health, and social participation. These activities include, but are not limited to, eating, personal hygiene (bathing), toileting, continence, dressing, and mobility (transferring).

### A. ADLs Analysis

Analyzing ADLs plays a crucial role in assessing an individual's physical vulnerability and functional autonomy. The literature identifies several benefits of ADL analysis for both individuals and healthcare providers:

- **Assessing independence**: Tools such as the Katz index [6] and the AGGIR scale [7] assess a person's ability to perform essential ADLs. Automatic human activity recognition can objectively evaluate an individual's independence, providing valuable insights for healthcare professionals and caregivers to tailor care plans.
- **Detecting frailty**: Frailty, characterized by an increased vulnerability to health problems, can be detected early by monitoring changes in ADLs [8]. This early detection enables proactive interventions to prevent further health decline.

- **Care planning**: The level of care required at home is often determined by an individual's ability to perform ADLs. Measuring ADLs assists in resource allocation and care planning, ensuring that necessary support is provided [9].
- **Improving quality of life**: By identifying difficulties through ADL analysis, targeted interventions or automated services can be deployed to enhance an individual's quality of life [10].

### B. ADLs Measurement

The advancement of AI technologies has enabled the automatic recognition of ADLs using data collected from a variety of connected devices in smart homes. Ambient Assisted Living (AAL) environments employ a range of environmental sensors, including acoustic, temperature, $CO_2$, humidity, power consumption, and passive infrared (PIR) sensors, as well as more advanced technologies like radar, lidar, and cameras [11].

Given the variability in ADL recognition algorithms due to differing sensor data sources, we propose an adaptive method capable of processing Boolean data from any sensor type. This approach includes a transformation layer that standardizes sensor readings into Boolean values by applying predefined thresholds, ensuring consistent input for ADL recognition algorithms and maintaining robust performance across different environments.

### C. Dataset

We utilized the Aruba dataset from the CASAS smart home project [12], a widely-used resource known for its extensive sensor coverage and detailed event logs. This dataset was collected over a period of seven months, from November 4, 2010, to June 30, 2011, capturing the daily activities of an elderly woman living alone with periodic family visits. The smart home was equipped with 31 wireless binary motion sensors, 5 temperature sensors, and 3 door sensors, strategically placed to monitor the resident's movements and interactions (Figure 1).

Each data entry includes a timestamp, sensor identifier, sensor status, and activity annotations, providing a rich data stream for robust activity recognition. With 1,719,557 recorded events, this dataset is one of the most comprehensive resources available for human activity recognition and has been widely used as a benchmark in the field. Its broad usage makes it ideal for comparing various algorithms and methodologies for activity recognition.

To enhance real-time processing while maintaining high recognition accuracy, we selected a subset of 11 sensors (motion and door sensors) specifically associated with the following activities: "Bed to toilet", "Eating", "Entering home", "Housework", "Leaving home", "Preparing meals", "Relaxing", "Sleeping", "Washing dishes", and "Working". This optimization aims to reduce processing overhead while still capturing key activity patterns. Figure 2 shows an example sequence of 'Sleeping' and 'Bed to Toilet' events recorded during the night of May 15, 2011, which reflects typical nighttime behavior in the dataset.

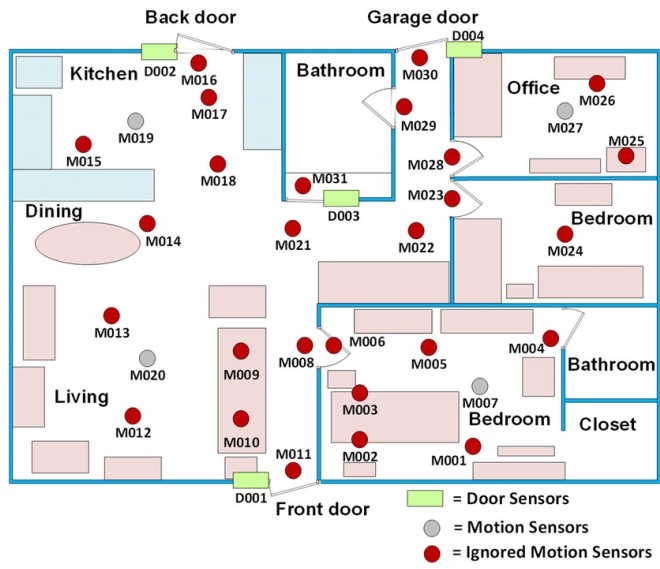

Fig. 1. Layout of the Aruba smart home.

```
Date        Time              Sensor  Status Annotation

2011-05-15  04:22:43.082116   M003    ON
2011-05-15  04:22:44.624551   M003    OFF   Sleeping end
2011-05-15  04:22:46.649038   M005    ON
2011-05-15  04:22:50.398987   M004    ON    Bed_to_Toilet begin
2011-05-15  04:22:50.472761   M005    OFF
2011-05-15  04:22:57.782528   T005    21
2011-05-15  04:22:58.533178   M004    OFF
2011-05-15  04:22:58.623508   M007    OFF
2011-05-15  04:25:44.619463   M004    ON
2011-05-15  04:25:45.993194   M007    ON
2011-05-15  04:25:50.590135   M004    OFF   Bed_to_Toilet end
2011-05-15  04:25:50.727788   M005    ON
2011-05-15  04:25:52.659384   M005    OFF
2011-05-15  04:25:55.446432   M003    ON    Sleeping begin
2011-05-15  04:25:57.258257   M007    OFF
```

Fig. 2. Sample of sensor data recorded in the Aruba Dataset.

### D. Literature review

We conducted a thorough literature review to analyze previous research in human activity recognition (HAR), focusing specifically on real-time recognition of Activities of Daily Living (ADLs). This review aims to present key methodologies and highlight both resolved challenges and areas requiring further investigation. The criteria for selecting relevant scientific articles included:

- **Real-time analysis**: Critical for detecting ADLs, real-time analysis enables immediate identification and monitoring, facilitating rapid responses in emergencies and continuous tracking of daily routines [13].
- **Ambient sensors**: These sensors, embedded in a person's environment (motion detectors, temperature sensors, door sensors, etc.), unobtrusively collect data on daily activities, playing a pivotal role in ADL recognition [14].
- **Spatiotemporal aspects**: HAR systems must account for both the location ('where') and timing ('when') of activi-

ties. Spatiotemporal modeling enhances activity detection by providing a contextual framework for interpreting sensor data [15].

- **Deep learning classifiers**: Unlike traditional methods, deep learning models such as CNNs and LSTMs can automatically extract relevant features from data, eliminating the need for manual feature design. These models are highly effective at capturing complex patterns and non-linear relationships, making them well-suited for ADLs, which vary greatly between individuals [16].

Table I summarizes the key methodologies from six approaches identified in our review, each chosen from a total of 3,222 relevant publications. These papers were selected based on the presence of real-time analysis, use of ambient sensors, spatiotemporal modeling, and classifier performance.

| References | Ambient sensors | Dynamic segmentation | Spatiotemporal | Classifier | Accuracy | Dataset |
|---|---|---|---|---|---|---|
| C. Chen et al. [17] | Yes | No | No | NB, MP, SMO, LB | - | CASAS |
| D. Liciotti et al. [18] | Yes | No | No | LSTM | 0.89 | CASAS |
| J. Wan et al. [19] | Yes | No | Yes | NB, BN, DT, NBT, HMM | 0.91 | CASAS |
| D. Bouchabou et al. [20] | Yes | No | No | LSTM | 0.84 | CASAS |
| H. Najeh et al. [2] | Yes | Yes | Yes | CNN2D | 0.81 | CASAS |
| Z. Xu et al. [3] | Yes | Yes | Yes | CNN-LSTM | 0.78 | CASAS |

TABLE I
COMPARATIVE ANALYSIS OF DIFFERENT APPROACHES IN ADL RECOGNITION

As seen in Table I, several approaches leverage deep learning and ambient sensors for ADL recognition, with varying degrees of success. Two specific approaches [2] and [3] were selected for further in-depth analysis and experimentation in our study due to their use of dynamic segmentation and spatiotemporal modeling.

## III. METHODOLOGY

Although the two selected methods focus on human activity recognition in smart home environments using dynamic segmentation and emergent modeling techniques, they present different methodological innovations and specific approaches. In [3], The authors present a model that uses spatio-temporal correlations to manage the segmentation of streaming data. This approach ensures that data segmentation is sensitive to both the location of sensor activations and the timing between these events, which helps accurately define each segment's context. This method uses stigmergy to build activity features explicitly represented as a network, aiding in

the contextual interpretation of sensor data without needing deep domain knowledge. In [2], the authors, inspired by the above work, emphasize a convolutional neural network (CNN) that is bootstrapped to dynamic segmentation combined with stigmergy-based encoding. Stigmergy is used here to create a feedback loop within the environment, enhancing the learning process by encoding and classifying segments of activities more efficiently. The approach strongly relies on the capabilities of CNNs to process and analyze the segmented data. The main difference between the two approaches is that Xu et al. employ stigmergy to construct activity features represented in a network model directly, whereas Najeh et al. use stigmergy to enhance the encoding process within a neural network framework. There is a difference in how stigmergy is applied—one is for building a self-updating model that adapts based on the activity context, and the other is for improving neural network training.

Both methods are based on the same principle but operate differently. They consist of two main phases: an offline phase for calculating and optimizing the parameters used to segment the real-time data stream, and an online phase for segmenting the real-time data stream, and then encoding and classifying the data segments to identify the specific activity being performed. The procedural steps for both phases are illustrated in Figure 3.

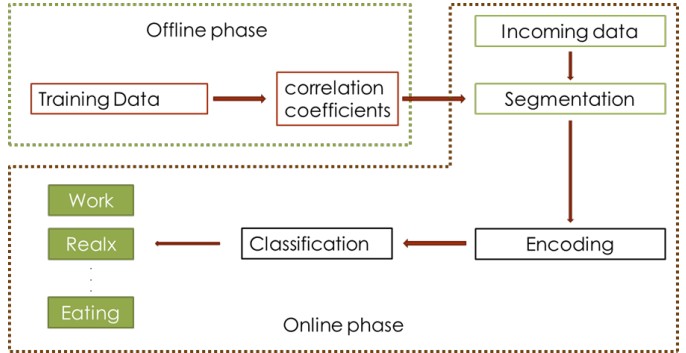

Fig. 3. ADL recognition methodology.

### A. Dataset overview

The dataset is structured to allow precise tracking and analysis of activities, providing detailed insights into the sequence and nature of captured events. It consists of three sets:

- **Events (E):** A set of individual events, each characterized by attributes such as date, time, sensor ID, and sensor value. Each event $E_i$ is a quadruplet of the form (date, time, sensorID, sensorValue).
- **Activity Segments (W):** This set consists of activity segments, where each segment $W_i$ is a sequence of events $E_i$. For example, in Figure 3, the "Bed to Toilet" segment captures the sequence of activities between the "Begin" and "End" states.

- **Sensor Identifiers (S):** A set of identifiers representing the various sensors deployed in the environment, ranging from $S_0$ to $S_{33}$, used to monitor and record activity data.

### B. Offline Phase

In the offline phase, we calculate spatial and temporal correlation coefficients to accurately segment sensor-derived events in real-time. These coefficients consider both the geographical context of sensor locations and the time intervals between events.

*1) Spatial Correlation:* Spatial correlation measures the relationships between sensor activations based on their spatial proximity. This is crucial for understanding how sensor events are interrelated within a given environment. We apply three distinct methods to calculate spatial correlation:

- **Pearson Product-Moment Correlation (PMC):** This method calculates the linear correlation between sensor events using covariance and standard deviations [2], as shown in Equation 1. It provides a straightforward measure of how sensor events vary together linearly.
- **Mutual Information (MI):** Mutual Information evaluates the dependency between sensors by assessing the probability distributions of their events [3]. It captures non-linear relationships between sensors by constructing a Sensor Correlation Matrix (SCM), as outlined in Equation 2. Significant correlations are identified using thresholds, with geographic proximity between sensors being a key factor.
- **Sequential Correlation Evaluation (SCE):** SCE enhances the SCM by considering the temporal sequence of sensor activations [19]. It evaluates correlations based on the order of sensor events, capturing both spatial and temporal dependencies (Equation 4).

$$\rho_{\mathbf{XY}} = \frac{\mathbf{cov}(\mathbf{X}, \mathbf{Y})}{\sigma_{\mathbf{X}} \ \sigma_{\mathbf{Y}}} \tag{1}$$

$$\mathrm{SCM}(s_i, s_j) = \frac{1}{|\mathbf{W}|} \sum_{k=1}^{|\mathbf{W}|} \theta(s_i, s_j) \tag{2}$$

$$\theta(s_i, s_j) = \begin{cases} 1, & \text{if } (s_i, s_j) \in \mathbf{W}_k \\ 0, & \text{else} \end{cases} \tag{3}$$

$$\mathrm{SCM}(s_i, s_j) = \frac{1}{|\mathbf{W}|} \sum_{p=1}^{|\mathbf{W}|} \sum_{k=1}^{|\mathbf{W}_p|-1} \gamma(s_k, s_i)\gamma(s_{k+1}, s_j) \tag{4}$$

$$\gamma(s_i, s_j) = \begin{cases} 1, & \text{if } s_p = s_i \\ 0, & \text{if } s_p \neq s_i \end{cases} \tag{5}$$

We have developed a modified approach to further enhance spatial correlation measurements. This modification accounts for the spatial separation of activity segments, improving the segmentation process. In Equation 6, the Sensor Correlation Matrix (SCM) values are adjusted based on sensor pairs from the first and last sensors of each activity, considering their relationships across all activities.

The following equation expresses the modified SCM calculation:

$$\begin{aligned} \mathrm{SCM}(s_i, s_j) = & \frac{1}{N} \sum_{k=1}^{m} \frac{1}{L_k} \sum_{(s,t) \in W_k} \big(\delta_{s,i}\delta_{t,j} + \delta_{s,j}\delta_{t,i}\big) \\ & - \frac{1}{F} \sum_{k=1}^{m} \sum_{(s,t) \in W_k} \bigg(\frac{1}{L_k} \sum_{u \in S \backslash W_k} \big(\delta_{s,i}\delta_{t,u} \\ & + \delta_{s,j}\delta_{t,u} + \delta_{u,i}\delta_{t,s} + \delta_{u,j}\delta_{t,s}\big)\bigg) \end{aligned} \tag{6}$$

where:

- $\mathrm{SCM}(s_i, s_j)$ represents the value of the SCM matrix for the pair $(s_i, s_j)$
- $N$ is the total number of activity segments
- $m$ is the number of distinct activities
- $L_k$ is the number of sensor pairs in the activity segment $W_k$
- $F$ is a normalization factor, for example $F = 4 \times m \times (n - L_k)$
- $\delta_{x,y}$ is the Kronecker delta function, defined as:

$$\delta_{x,y} = \begin{cases} 1 & \text{if } x = y \\ 0 & \text{otherwise} \end{cases}$$

- $S$ is the set of sensor identifiers.

This enhanced SCM calculation provides a more accurate representation of sensor relationships across activity segments. The results of the Sensor Correlation Matrix are displayed in Figure 4.

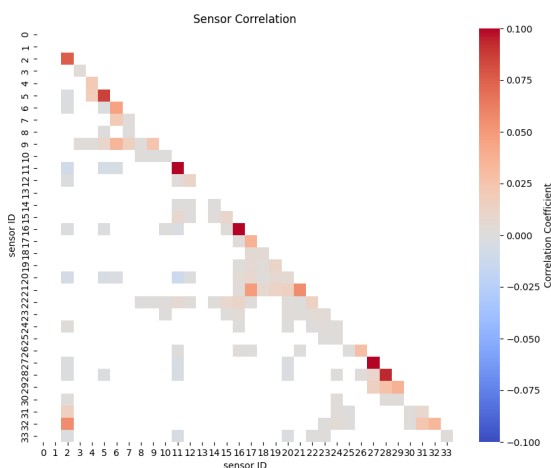

Fig. 4. Sensor Correlation Values in the SCM Matrix.

*2) Time Correlation:* Time correlation ensures that events are grouped based on both spatial and temporal proximity. We use two thresholds to determine whether an event belongs to a segment:

- MTI (Equation 7) measures the maximum time between consecutive sensor events within an activity.
- MTS (Equation 8) represents the total time span from the start to the end of an activity segment.

$$\text{MTI}(s_i, s_j) = \max_{p=1}^{|\omega_k|-1} (T_{p+1} - T_p)\, \delta_{p,i}\delta_{p+1,j} \tag{7}$$

$$\text{MTS}(s_i, s_j) = \max_{k=1}^{|\omega|}(T_{\text{last},j} - T_{\text{first},i}) \tag{8}$$

### C. Online Phase

In the online phase, the algorithm processes incoming sensor data to dynamically segment events based on spatial and temporal correlation conditions. After segmentation, the next step involves encoding and classifying the data to recognize specific activities.

*1) Segmentation:* The dynamic segmentation process groups sensor events based on four key parameters: the Sensor Correlation Matrix (SCM), Sensor Correlation Threshold (SCT), Maximum Time Interval (MTI), and Maximum Time Span (MTS). Algorithm 1 outlines the steps of this process.

Algorithm 1 provides the detailed steps involved in this dynamic segmentation process.

---

**Algorithm 1** Dynamic Sensor Data Segmentation Algorithm.

    **Require:** Streaming sensor data: $E = (E_1, E_2, \ldots, E_n)$
    **Require:** Initial segmentation window: $W_1 = \{\}$
    **Ensure:** Segments: $W_1 = \{E_1, E_2, \ldots, E_k\}$
    **Conditions:** SCM/SCT, MTI and MTS
    **While** incoming sensor event **do**
        $add = \text{False}$
        **if** $W_i$ is empty or $add = \text{False}$ **then**
            $W_1 = \{E_i\}$
            $E_{\text{first}} = E_i$
            $E_{\text{last}} = E_i$
        **else**
            $T_{\text{cor\_last}} = (T_i - T_{\text{last}}) > MTI(S_i, S_{\text{last}})$
            $T_{\text{cor\_first}} = (T_i - T_{\text{first}}) > MTS(S_i, S_{\text{last}})$
            $S_{\text{cor}} = SCM(S_i, S_j) \geq SCT$
            **if** $S_{\text{cor}}$ and $T_{\text{cor\_first}}$ and $T_{\text{cor\_last}}$ **then**
                $W_1 = W_i + \{E_i\}$
            **else**
                $add = \text{True}$

---

This algorithm continuously processes sensor events to determine whether they should be added to the current segment based on spatial and temporal correlations. If the conditions (SCM, SCT, MTI, MTS) are satisfied, the event is included in the ongoing segment; otherwise, a new segment is initiated. This approach ensures that events with strong spatial and temporal relationships are grouped, which is essential for accurate activity recognition.

*2) Encoding:* The encoding process transforms the input segment into a latent space representation using an adjacency matrix. This matrix captures the relationships between sensors during each segment. The transformation is achieved by first constructing a directed network of sensor events, followed by calculating the weight of each sensor state change using the Directed Weight Network (DWN) approach [3].

Each weight reflects the duration of a sensor's activation and is stored in a matrix for further processing, as shown in Figure 5. The encoding process consists of the following steps:

- **Step 1: Segment representation.** Part (a) in Figure 5 illustrates the events forming a segment. The first sensor in the DWN corresponds to the last sensor activated in the previous segment. This ensures continuity between consecutive segments.
- **Step 2: Directed weight calculation.** The active sensors in the current segment (those with SensorValue = ON) are used to compute the weights. The weight for each sensor state change is calculated based on the sensor's activation duration, following the formula in Equation 9.

$$I = \frac{(1 - \rho)^{(T_e - t_e)} - (1 - \rho)^{(T_e - t_s)}}{\rho} \tag{9}$$

where:
- – $T_e$ is the time of the last event in the segment,
- – $t_e$ and $t_s$ are the activation and deactivation times of the sensors,
- – $\rho$ is a decay factor set to 0.2 as per [2].

- **Step 3: Adjacency matrix construction.** The weights calculated in Step 2 are stored in an adjacency matrix. This matrix has dimensions corresponding to the number of sensors in the environment, and it encodes the relationships between sensors within the segment. Part (d) in Figure 5 demonstrates how the directed weights are integrated into the matrix.

This encoding process efficiently captures both the temporal and spatial dependencies of sensor events, enabling accurate activity recognition in subsequent stages.

*3) Classification:* Activity recognition classification assigns labels to temporal sequences based on observed behaviors. This is primarily achieved using deep learning models that analyze patterns from sensor data. By training on annotated datasets, the model predicts activities in real time by mapping data sequences to specific labels. Our classifier, which processes input as two-dimensional matrices, leverages Convolutional Neural Networks (CNNs) designed for 2D data (CNN2D). These networks are well-suited for this task due to their ability to capture spatial relationships, efficiently share weights across layers, and prioritize key features [21].

The CNN architecture used incorporates several specialized layers to optimize performance. Batch Normalization is applied to accelerate convergence during training, stabilize gradients, and reduce the risk of overfitting. A Flatten layer converts the 2D matrix data into a one-dimensional vector, preparing it for input into the fully connected Dense layer,

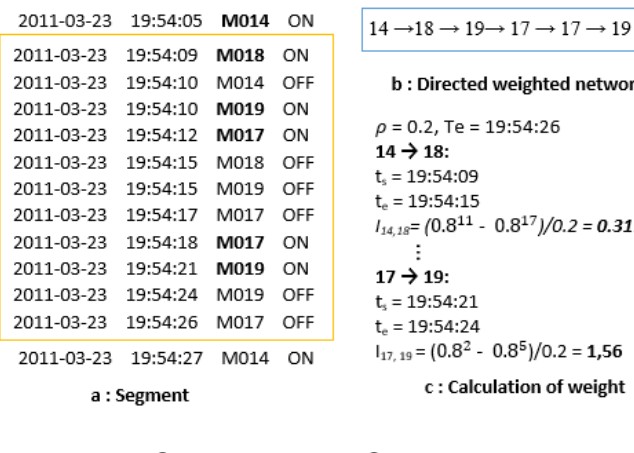

Fig. 5. Segment encoding using the Directed Weight Network (DWN).

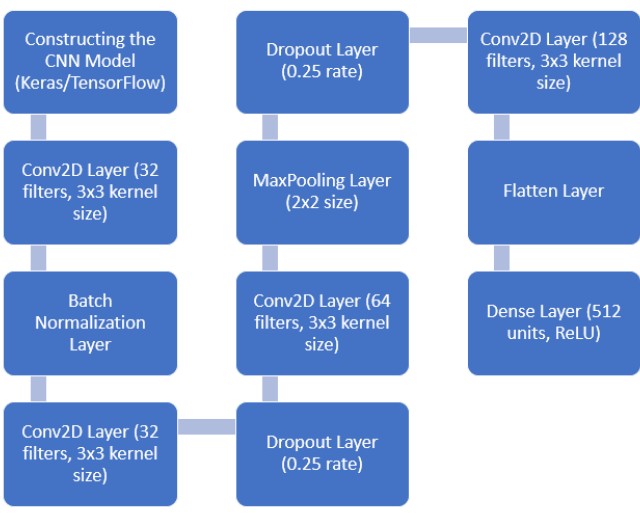

Fig. 6. CNN Architecture for Activity Recognition.

which computes predictions based on a linear combination of inputs with an activation function. To further enhance generalization, regularization techniques such as Dropout and MaxPooling2D are employed. Dropout randomly excludes subsets of features during training to prevent overfitting, while MaxPooling2D reduces the spatial dimensions, allowing the model to retain the most important features from the input data.

Figure 6 illustrates the architecture and configuration of the layers used in our CNN model.

## IV. PERFORMANCE EVALUATION

We evaluate the model's classification performance using key metrics: accuracy, precision, recall, and the F1-score. These metrics provide a comprehensive understanding of the model's effectiveness, especially in situations where class imbalance is present.

- **Accuracy** quantifies the proportion of correct predictions, both positive and negative:

$$\text{Accuracy} = \frac{TP + TN}{TP + TN + FP + FN} \quad (10)$$

Although accuracy is a useful general indicator, it can be misleading in cases of imbalanced data distributions.

- **Precision (P)** measures the proportion of correctly predicted positives out of all positive predictions:

$$P = \frac{TP}{TP + FP} \quad (11)$$

- **Recall (R)**, or sensitivity, evaluates the model's ability to correctly identify all actual positives:

$$R = \frac{TP}{TP + FN} \quad (12)$$

- **F1-score** provides a balanced measure between precision and recall:

$$F1 = 2 \times \left( \frac{\text{Precision} \times \text{Recall}}{\text{Precision} + \text{Recall}} \right) \quad (13)$$

In the online dynamic segmentation task, a key challenge is the potential discrepancy between the number of predicted segments and the actual segments. To address this issue, we propose using sequence alignment techniques to assess the similarity between the predicted and actual event sequences. Specifically, we employ the Needleman-Wunsch algorithm for global alignment, which accounts for matches, substitutions, and insertions/deletions (indels) between sequences.

The Needleman-Wunsch algorithm initializes a score matrix $S$ with dimensions $(n + 1) \times (m + 1)$, where $n$ and $m$ represent the lengths of the actual and predicted label sequences, respectively. The alignment process is guided by maximizing match scores or applying penalties for gaps and mismatches:

$$S[i][j] = \max \begin{cases} S[i-1][j-1] + M(P[i], R[j]) \\ S[i-1][j] - d \\ S[i][j-1] - d \end{cases} \quad (14)$$

Here, $M(P[i], R[j])$ assigns a score of +1 for a match and $-d$ for a mismatch.

Once the score matrix is computed, backtracking is performed to find the optimal alignment, considering matches, insertions, and deletions. The alignment accuracy is then computed as the percentage of exact matches:

$$\text{Accuracy} = \frac{C}{n} \times 100 \quad (15)$$

where $C$ represents the number of exact matches.

## V. EXPERIMENTS AND RESULTS

To evaluate the accuracy of the recognition algorithm in isolation, the system was first tested without the segmentation step. The model was trained on 80% of the Aruba dataset and tested on the remaining 20% annotated segments. The results demonstrated a high accuracy of 98.3% and an F1-score of 0.982, indicating that the classification model is highly effective in terms of both precision and recall.

Figure 7 presents the classification results, including the confusion matrix and accuracy percentages for all activities.

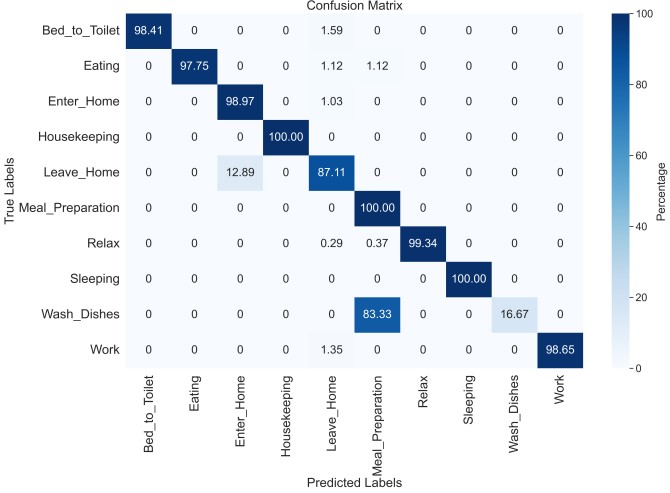

Fig. 7.  Confusion Matrix and Accuracy for All Activities.

While the overall classification performance is strong, the system encounters difficulties in distinguishing between similar activities such as 'Enter_Home' and 'Leave_Home' or 'Meal_Preparation' and 'Wash_Dishes'. These challenges arise due to similarities in the sensor activations triggered by these activities, which results in overlapping features and sensor patterns.

In evaluating the complete system, which includes dynamic segmentation and activity recognition, the model was again trained on 80% of the Aruba dataset and tested on the remaining 20%. The segmentation process generated 1629 segments, slightly more than the actual 1611 segments, due to variability in task execution. Dynamic segmentation may produce segments that do not perfectly align with actual activity boundaries, leading to discrepancies in the predicted number of activities. Consequently, confusion matrices are not always suitable for evaluating model performance. To address this, we employed the Needleman-Wunsch algorithm for global alignment between predicted and actual activity sequences, accounting for matches, substitutions, and insertions/deletions (indels) between sequences.

Figure 8 illustrates the output of this method, offering a more consistent evaluation of the activity recognition process.

Of the 1629 predicted activities, 1340 corresponded to actual activities, resulting in an overall accuracy rate of 83.2%. This accuracy demonstrates the model's capacity to handle the

**Global alignment (Needleman-Wunsch)**

```
Actual activities

6 6 6 6 6 8 0 0 8 5 5 6 4 2 5 5 5 4 2 6 6 6 6 5 6 5 6 5 5 6 6 6 -
      | | |   | | | | | | | | |   | | | |   | | | | | | | | |
- 5 6 6 6 1 0 8 5 5 6 4 2 5 5 5 8 2 6 6 6 5 5 6 5 6 5 5 6 6 6 8

Predicted activities
```

Fig. 8.  Global Alignment of Actual and Predicted Activities via the Needleman-Wunsch Method.

complexity of dynamic segmentation and online recognition in smart home environments.

Table II presents a comparative analysis of our method's accuracy relative to similar approaches that leverage spatiotemporal data and real-time segmentation. Our method demonstrates a significant improvement, particularly in its ability to segment real-time data and recognize activities accurately.

| Approach | Ambient sensors | Dynamic segmentation | Spatiotemporal | Classifier | Accuracy | Dataset |
|---|---|---|---|---|---|---|
| Z. Xu et al. [3] | Yes | Yes | Yes | CNN-LSTM | 0.78 | CASAS |
| H. Najeh et al. [2] | Yes | Yes | Yes | CNN2D | 0.81 | CASAS |
| Our method | Yes | Yes | Yes | CNN2D | **0.83** | CASAS |

TABLE II
COMPARATIVE ANALYSIS OF DIFFERENT APPROACHES IN ADL RECOGNITION.

## VI. CONCLUSION

This research presents a robust method for real-time ADL recognition using IoT data. The results confirm the effectiveness and reliability of our approach, particularly when compared to state-of-the-art dynamic segmentation methods leveraging spatiotemporal data from IoT sensors. An essential contribution of our work is the improved segmentation process, which enhances the system's ability to detect activities by globally aligning sequences of events accurately.

In addition to improving segmentation, computational efficiency is critical for real-time human activity recognition (HAR), especially on embedded devices. Building on the work of Najeh et al. [24], who optimized HAR algorithms for resource-constrained hardware, we anticipate that our system can maintain high accuracy on low-power, low-cost devices. However, our approach differentiates itself by incorporating an additional layer dedicated to analyzing ADL sequences using

the Needleman-Wunsch algorithm. This layer is optimized to ensure compatibility with embedded systems that have limited resources. Further testing across various hardware and software configurations will allow us to validate our approach fully.

During our experiments, we encountered occasional difficulties in distinguishing between very similar activities. The literature addresses this challenge through advanced feature extraction (e.g., temporal, frequency-based, and spatiotemporal features), deep learning approaches (e.g., CNNs, RNNs, and transformers), and multi-modal sensor fusion. Combining these techniques improves the system's ability to differentiate similar activities. Our hybrid approach integrates spatiotemporal features, deep learning models, multi-modal sensor data fusion, and contextual segmentation, all tailored to individual user behavior. Looking ahead, we plan to explore the incorporation of additional sensors (e.g., entry/exit points) and more contextual information, such as activity duration and sensor-activity signatures. However, these additions will involve additional data collection and integration costs, which must be carefully evaluated.

Finally, it is essential to emphasize that the spatial arrangement of sensors during installation significantly impacts performance. In the future, we plan to develop a sensor installation simulator to optimize sensor placement, improving the accuracy and reliability of ADL detection across various smart home configurations.

## ACKNOWLEDGMENT

This work was conducted within the framework of the *Chaire Maintien à Domicile* (M@D), supported by the foundations of the University of Southern Brittany. It is also part of operation 4.2.7: *Measurement and Analysis of Activities of Daily Living (ADL)*, as part of the *Handicap Innovation Territoire (HIT)* project, supported by Lorient Agglomeration. We would like to extend our thanks to Ndeye-Maguette Diagne, an intern at the *Chaire M@D*, for her invaluable contribution to this project and to the development of part of this work.

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
