# OpenReview forum: "Enhanced Online Segmentation and Performance Evaluation Method for Real-Time Activity Recognition in Smart Homes"
_IEEE.org/EMBS/BHI/2024/Conference — IEEE BHI'24_

### Official Review · Reviewer_1brP · 2024-08-09
**Assessing the Efficacy and Innovation in Real-Time Activity Recognition: A Review of Enhanced Online Segmentation Methods for Smart Homes**

**Overall Rating:** 7
**Confidence:** 4

**Other Quality Metrics:**

Clarity of Writing: Great
Clinical Significance: Fair
Methodological Novelty: Great
Experiments and Results: Good

**Questions For The Authors:**

1. Have you tested your method on datasets other than the Aruba dataset? If not, do you plan to evaluate your approach on other datasets with different sensor configurations or activity patterns to assess its generalizability?
2. Have you considered testing your method in a live smart home environment with actual residents performing daily activities? If so, what were the results, and how did the method perform under real-world conditions?
3. While the Needleman-Wunsch method is used for evaluating activity alignment, have you considered other evaluation metrics or techniques, such as sequence-to-sequence learning models, that might provide additional insights into your method’s performance?
4. The performance of your method is sensitive to sensor placement. How do you plan to address variations in sensor placement in different homes? Could a self-adapting algorithm or a more flexible segmentation method mitigate this issue?

**Strengths:**

1. The paper introduces an enhanced method for real-time activity recognition in smart homes, combining improved online segmentation with a novel performance evaluation technique.
2. It presents a detailed, two-phase approach (offline and online) that addresses both the segmentation and classification aspects of activity recognition.
3. Their method leverages spatiotemporal features of IoT sensors to enhance the segmentation process, which is a key improvement over existing methods.
4. The use of the Needleman-Wunsch method for evaluating predicted activities is a creative solution to the challenge of misaligned predictions in real-time systems. The method achieves an impressive accuracy rate of 83.2% in real-time activity recognition. This performance is superior to many existing methods, particularly in handling the complexity of spatiotemporal data in smart homes.

**Summary Of The Paper:**

The paper introduces a robust method for real-time recognition of Activities of Daily Living (ADLs) using IoT data in smart home environments. The approach enhances online segmentation by exploiting spatiotemporal features of IoT sensors and employs the Needleman-Wunsch method for evaluating predicted activities. The method addresses discrepancies between predicted and actual activities in real-time recognition scenarios. The authors used the Aruba dataset from the CASAS smart home project, which provided comprehensive data for training and evaluating their model. The proposed method involves two phases:

1. Offline Phase: In this phase, spatial and temporal correlation coefficients are computed to segment sensor-derived events accurately. The study explores various methods, including Pearson Product Moment Correlation, Mutual Information, and Sequential Correlation Evaluation.
2. Online Phase: The incoming sensor data is segmented dynamically using the calculated spatial and temporal correlation conditions. The data is then encoded and classified using a Convolutional Neural Network (CNN2D) designed for 2D data.

Their method achieved an accuracy rate of 83.2% for activity recognition. The performance was benchmarked against other methods using similar datasets, and it demonstrated significant improvements. The method showed high accuracy in classifying activities, although some activities, such as 'Enter Home' and 'Leave Home,' posed challenges due to similarities in sensor activations.

In conclusion, the paper presents a robust and reliable method for real-time ADL recognition using IoT data in smart homes. The proposed approach enhances segmentation accuracy and provides a novel evaluation methodology to address discrepancies in predicted activities, making it a significant contribution to the field of Ambient Assisted Living (AAL). The method outperforms existing dynamic segmentation methods and paves the way for future improvements through sensor installation simulations.

**Weaknesses:**

1. The study relies solely on the Aruba dataset. While this dataset is well-known, using multiple datasets or validating the method across different smart home environments could provide more robust evidence of the method's generalizability. It would've been insightful to test in a real-time, real-world environment. The paper doesn't thoroughly address the computational requirements of the method, which is crucial for real-time applications in resource-constrained IoT environments.
2. There's a significant drop in accuracy from 98.3% (without segmentation) to 83.2% (with dynamic segmentation). While this is still an improvement over existing methods, the paper could benefit from a more detailed analysis of this performance gap.
3. The paper doesn't discuss how well the method scales with an increasing number of sensors or activities, which is important for larger or more complex smart home setups.

---

### Official Review · Reviewer_KN7Q · 2024-08-10
**Enhanced Online Segmentation and Performance Evaluation Method for Real-Time Activity Recognition in Smart Homes**

**Overall Rating:** 6
**Confidence:** 4

**Other Quality Metrics:**

(a) Clarity of writing; Fair
(b) Clinical Significance; Poor
(c) Methodological Novelty; Fair
(d) Experiments and Results; Fair

**Questions For The Authors:**

Optimize Computational Efficiency: Provide a detailed analysis of the computational requirements and explore optimization techniques to ensure the method can be implemented effectively in real-time scenarios.
Enhance Activity Differentiation: Investigate additional features or methods to better differentiate between similar activities, improving the system's overall accuracy and reliability.
Assess Real-time Performance: Conduct real-time performance tests to evaluate the system's latency and responsiveness, ensuring it meets the demands of practical applications in smart home environments.

**Strengths:**

Innovative Methodology: The combination of spatiotemporal features and the Needleman-Wunsch method for activity recognition is an innovative approach that enhances segmentation accuracy and overall system performance.
Practical Application: The focus on real-time recognition of ADLs in smart homes addresses a crucial need in Ambient Assisted Living (AAL) environments, especially for the elderly and disabled.
Comprehensive Evaluation: The use of the Aruba dataset and various performance metrics provides a thorough evaluation of the proposed method, demonstrating its effectiveness in a real-world context.
Future-Oriented: The plan to develop a sensor installation simulator for optimizing the accuracy and reliability of the system shows a forward-thinking approach to continuous improvement.

**Summary Of The Paper:**

This paper presents an enhanced method for real-time recognition of Activities of Daily Living (ADLs) using IoT data in smart home environments. The approach focuses on improving online segmentation by utilizing spatiotemporal features from IoT sensors and employing the Needleman-Wunsch method to evaluate the alignment between predicted and actual activities. The method was tested using the Aruba dataset, achieving an accuracy of 83.2%, which outperforms existing dynamic segmentation methods. The paper highlights the importance of accurate segmentation and activity recognition in smart homes for elderly and disabled individuals, aiming to improve personal assistance and health monitoring. Future work includes the development of a sensor installation simulator to further enhance the system's accuracy and reliability.

**Weaknesses:**

Complexity of Implementation: The proposed method, particularly the use of the Needleman-Wunsch algorithm, may add complexity to the system, potentially affecting its real-time performance. A discussion on the computational overhead and strategies to mitigate it would be beneficial.
Handling of Similar Activities: The paper notes difficulties in distinguishing between similar activities such as 'Enter Home' and 'Leave Home.' Further refinement in distinguishing such activities, possibly through additional context-aware features, would improve the system's robustness.
Real-time Performance Evaluation: While the accuracy of the method is reported, there is limited discussion on the real-time performance and latency of the system. Understanding the system's responsiveness in real-time applications is crucial for its practical deployment.

---

### Official Review · Reviewer_kLK7 · 2024-08-12
**Enhanced Online Segmentation and Performance Evaluation Method for Real-Time Activity Recognition in Smart Homes**

**Overall Rating:** 5
**Confidence:** 2

**Other Quality Metrics:**

(a) Clarity of writing: good
(b) Clinical Significance: poor
(c) Methodological Novelty: fair
 (d) Experiments and Results: good

**Questions For The Authors:**

no

**Strengths:**

The proposed approach enhanced online segmentation by exploiting spatiotemporal features of IoT sensors and employs the Needleman-Wunsch method for evaluating predicted activities, addressing discrepancies between predicted and actual activities.

**Summary Of The Paper:**

This paper introduced a robust method for real-time recognition of Activities of Daily Living (ADLs) using IoT data in smart home environments.

**Weaknesses:**

no

---

### Decision · Program_Chairs · 2024-09-23

Accept